# Inherently Faithful Attention Maps for Vision Transformers

## Abstract

We introduce an attention-based method that uses learned binary attention masks to ensure that only attended image regions influence the prediction, a property we term **inherently faithful**. Context can strongly affect object perception, sometimes leading to biased representations, particularly when objects appear in out-of-distribution backgrounds. At the same time, many image-level object-centric tasks require identifying relevant regions, often requiring context. To address this conundrum, we propose a two-stage framework: stage 1 processes the full image to discover object parts and identify task-relevant regions, while stage 2 leverages input attention masking to restrict its receptive field to these regions, enabling a focused analysis while filtering out potentially spurious information. Both stages are trained jointly, allowing stage 2 to refine stage 1. Extensive experiments across diverse benchmarks demonstrate that our approach significantly improves robustness against spurious correlations and out-of-distribution backgrounds. Code is available in this anonymized repository.

## 1 Introduction

Deep Learning (DL) models often rely on contextual cues to learn object representations. While this can be beneficial for certain tasks, it can also introduce spurious correlations on which the model learns to rely, hampering generalization Rosenfeld et al. (2018); Choi et al. (2012); Xiao et al. (2021). A common example is when models prioritize background cues over intrinsic object properties, leading to failures in out-of-distribution (OOD) settings where such correlations no longer hold Beery et al. (2018); Aniraj et al. (2023). It is therefore crucial to ensure that the model focuses on task-relevant image regions and that users can assess whether the attended regions are appropriate.

To obtain these insights, many *post hoc* explainability methods Minh et al. (2022) have been proposed, commonly categorized as eXplainable AI (XAI) tools, which generate explanations in the form of saliency maps, providing a glimpse into the model's decision-making process without altering its structure. While *post hoc* methods are appealing because they do not affect model performance, this also means that they are unsuitable to prevent the model from latching onto spurious cues. Additionally, these methods offer no guarantee that the explanations are faithful to the model's reasoning Adebayo et al. (2018); Feng et al. (2018); Friedman et al. (2023), making failures difficult to detect Bove et al. (2024) and potentially misleading users Rudin (2019).

In contrast, models that integrate spatial attention maps directly into their inference process can help guiding the model towards focusing on the correct image regions and have the potential to provide guarantees of faithfulness, as they reveal the reasoning of the model rather than relying on a *post hoc* approximation. Among these, part discovery methods Huang & Li (2020); van der Klis et al. (2023); Aniraj et al. (2024) have gained prominence for inherently highlighting relevant object parts through learned attention maps. These methods typically compute the similarity between learned prototypes and high-level feature representations, using the resulting soft attention maps to assign greater importance to specific regions when forming the final image representation.

However, we argue that the attention maps produced by such methods do not fully capture the model's reasoning, leading to the same reliability issues as post hoc approaches. Specifically, (i) high-level feature

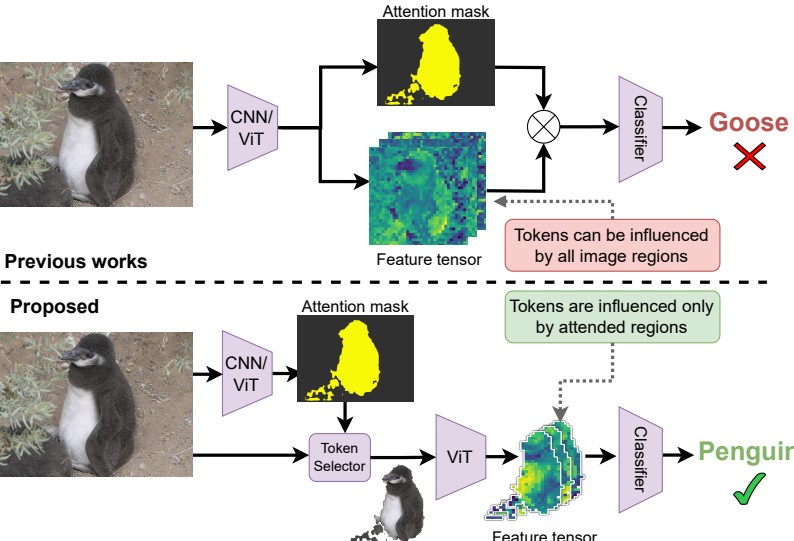

Figure 1: Previous attention-based approaches apply the attention mask to a deep feature tensor, where all locations can be affected by the whole image due to large receptive fields (top). Our approach ensures that only the selected tokens contribute to the downstream task (bottom).

representations at later stages aggregate information from the entire image due to their large receptive field, resulting in unintended background dependence; and (ii) soft attention masks, being non-binary, assign non-zero weights to all locations, allowing further unintended information leakage.

To address these issues, we propose a two-step framework that jointly learns a region selector and a Vision Transformer (ViT)-based classification model, where the latter relies solely on the selected image regions (Fig. 1). Building on a recent part discovery method Aniraj et al. (2024), we use discretized attention maps—formed by merging discovered parts—to explicitly select image regions for a second-stage classifier. This classifier, which also takes the raw image as input, has only access to the selected regions, thus mitigating spurious correlations present in other regions. Our approach provides an end-to-end signal that jointly optimizes both stages. Thus, *our core contribution is a model that explicitly ignores image regions that do not contribute to its prediction, ensuring robustness against spurious correlations present in those regions.* This design allows for systematic evaluation using established benchmarks for robustness against spurious correlations.

## 2 Related Works

**Spatial attention in computer vision.** Attention mechanisms induce the model to focus on a subset of the input that is deemed relevant to solve the task at hand. Originally devised as a means to reduce computational load in image classification Mnih et al. (2014), spatial attention mechanisms started to gain popularity for tasks such as captioning Xu et al. (2015), visual reasoning Hudson & Manning (2018), and other tasks Guo et al. (2022) where a sharp focus on a sequence of relevant image regions allows the model to decompose the complex task into multiple, simpler ones. Recent work on part discovery Huang & Li (2020); van der Klis et al. (2023); Aniraj et al. (2024) also leverages attention mechanisms. These approaches assume that focusing the attention on the correct parts will lead to better classification results, and leverage this learning signal to discover the semantic parts that compose the objects of interest. However, all of these methods apply attention to deep feature representations, where large receptive fields allow regions outside the attended area to influence the attended regions. This can potentially reduce *faithfulness*, or how well the attention map actually coincides with the image regions that matter for the downstream task. This has

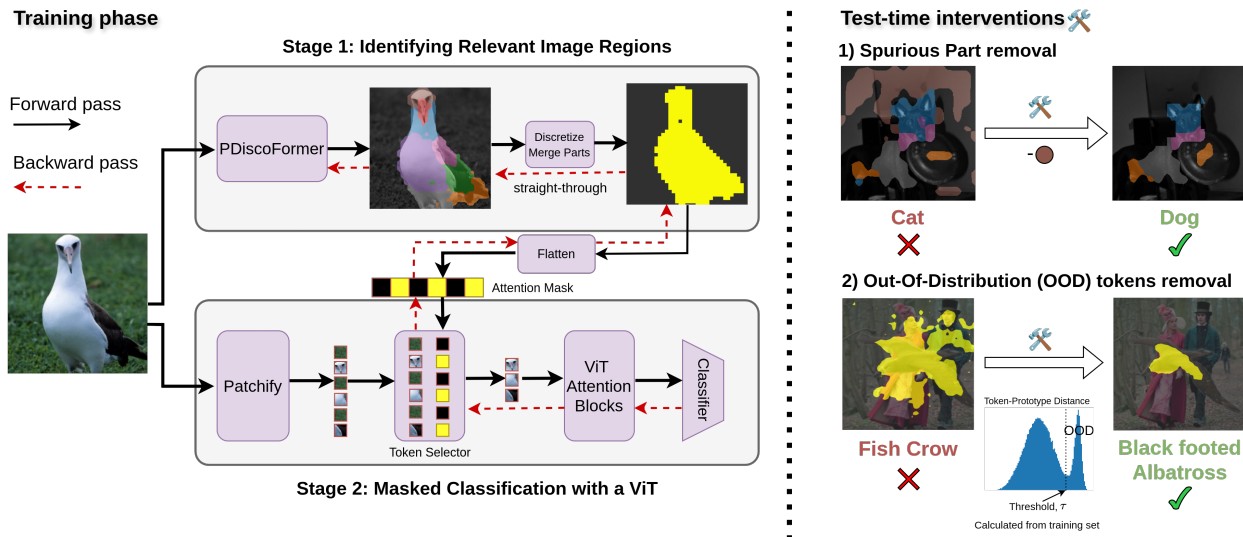

Figure 2: **Left:** iFAM first discovers task-relevant regions (Stage 1) and then classifies using only the selected regions (Stage 2), preventing reliance on background cues. **Right:** At test time, we leverage the model's inherently faithful region attribution to design (training-free) intervention strategies that further enhance robustness to spurious correlations.

led to work aiming at measuring the faithfulness of attention maps in ViTs Wu et al. (2024b), as well as to methods improving it Xie et al. (2022); Wu et al. (2024a); Ntrougkas et al. (2024) Unlike these works, our two-stage framework ensures that the attention maps are inherently faithful by explicitly constraining the predictor's receptive field.

**Local object representations.** Object-centric computer vision tasks require representations that remain invariant to changes in backgrounds and co-occurring objects. Previous works provide local object representations via mask-invariance losses Stone et al. (2017), clustering-like losses Yun et al. (2022) or directly altering the attention mechanism Ibtehaz et al. (2024). While some methods aim to align post-hoc explanations with segmentation maps Ross et al. (2017), they do not guarantee that only attended areas contribute to the decision, with studies highlighting information contamination from outside the object attention masks due to large receptive fields Aniraj et al. (2023).

**Input attention maps for interpretability.** Auxiliary mask predictors have been proposed to explain black-box classifiers by identifying minimal masks that preserve predictions without retraining Yuan et al. (2020); Phang et al. (2020); Stalder et al. (2022); Brinner & Zarrieß (2023); Nalmpantis et al. (2023); Zhang et al. (2024). Others use *post hoc* attribution maps to guide training Ismail et al. (2021). Closer to our approach, joint amortized explanation methods (JAMs) Chen et al. (2018); Yoon et al. (2018); Ganjdanesh et al. (2022) jointly learn selector and predictor models but risk encoding class information through the selection pattern Jethani et al. (2021); Puli et al. (2024). Although more recent methods have proposed solutions to alleviate this drawback, they involve either unstructured selection masks Jethani et al. (2021) or simplistic ones parametrized as a single spatial Gaussian Ganjdanesh et al. (2022). COMET Zhang et al. (2024) takes a step further and aims at finding the complete foreground, rather than a sufficient mask. In contrast to these works, our approach introduces a mechanism specifically developed for ViTs and leverages recent advances in part discovery to provide a rich spatial representation to the predictor. Empirical results show this improves performance, particularly in the presence of spurious cues.

**Input attention maps for robustness.** Joint learning of input masks has also been explored to enhance model robustness. Xiang et al. (2021) shows that limiting the receptive field and applying targeted patch masking improves adversarial robustness. Spurious correlations can be mitigated by isolating foreground regions and constructing image composites with mismatched backgrounds Xiao et al. (2023); Noohdani et al. (2024); Chakraborty et al. (2024), encouraging the model to rely on foreground cues. Asgari et al. (2022)

masks key image regions using attribution maps, forcing the model to identify alternative features and assess potential spurious correlations. Multiple spurious cues can coexist in a dataset, and techniques designed to mitigate one may inadvertently amplify another Li et al. (2023). In this work, we leverage the part discovery mechanism to simultaneously model several of these correlations.

**Relationship to Token Pruning and Sparse Attention.** Our approach is distinct from efficiency-focused methods like sparse attention Wei et al. (2023); Zhu et al. (2023) or token pruning Tang et al. (2023); Rao et al. (2021). While these methods also select a subset of tokens for processing, their primary objective is to accelerate inference. The token selection criteria are often based on discriminativeness for the downstream task. In contrast, iFAM's token selection is driven by a joint objective that combines classification with part-shaping losses to discover semantically consistent foreground regions.

## 3 Methodology

**iFAM** (**I**nherently **F**aithful **A**ttention **M**aps for vision transformers) depicted in Fig. 2, consists of two stages: the first one has access to the whole image and predicts which image regions should be selected for the second stage. These selected regions then define the receptive field used by the second stage for solving the downstream task. This design ensures that the second stage can only pay attention to the selected image regions, guaranteeing that it cannot make use of any information outside the mask.

### 3.1 Early vs Late Masking

**Existing attention-based methods** learn two functions on the input: a selector $f_{\text{sel}}$, with $\mathbf{s} = f_{\text{sel}}(\mathbf{x})$, and a feature extractor $f_{\text{pred}}$, with $\mathbf{h} = f_{\text{pred}}(\mathbf{x})$. The input $\mathbf{x} \in \mathbb{R}^{D_{\text{in}} \times N}$ is a set of $N$ elements, such as pixels or tokens, $\mathbf{h} \in \mathbb{R}^{D_{\text{out}} \times N}$ is a set of feature vectors and $\mathbf{s} \in \{0, 1\}^N$ is a binary selection mask[1]. An image feature vector $\mathbf{z} \in \mathbb{R}^{D_{\text{out}}}$, to be used for some downstream task, is then computed as:

$$\mathbf{z} = m(f_{\text{pred}}(\mathbf{x}), f_{\text{sel}}(\mathbf{x})), \tag{1}$$

where $m(\cdot, \cdot)$ is some masking and aggregator function. A common choice is a weighted average:

$$\mathbf{z} = \frac{1}{N} \sum_{i=1}^{N} s_i \mathbf{h}_i. \tag{2}$$

**With our approach**, the image feature vector is computed by applying the selector (stage-1) and the feature extractor (stage-2) sequentially:

$$\mathbf{z} = f_{\text{pred}}(m(\mathbf{x}, f_{\text{sel}}(\mathbf{x}))), \tag{3}$$

where $m(\cdot, \cdot)$ is now a masking function applied to the input of $f_{\text{pred}}$, and the aggregation is assumed to be performed within $f_{\text{pred}}$. Since the masking happens at the input level, the receptive field is determined by the mask for any aggregation method.

**Implementation on a ViT with attention masks.** In the case the model $f_{\text{pred}}$ is based on self-attention Vaswani et al. (2017), such as a ViT, $m(\cdot, \cdot)$ can be implemented by modulating the self-attention in each layer with a mask $\mathbf{M} \in \mathbb{R}^{N \times N}$:

$$\text{Attention}(\mathbf{Q}, \mathbf{K}, \mathbf{V}) = \text{softmax}\left(\frac{\mathbf{Q}\mathbf{K}^\top}{\sqrt{D}} + \mathbf{M}\right)\mathbf{V}, \tag{4}$$

where the elements in $\mathbf{M}$ are defined as:

$$M_{ij} = \begin{cases} -\infty, & \text{if } s_i = 0 \text{ or } s_j = 0 \\ 0, & \text{otherwise.} \end{cases} \tag{5}$$

This forces the attention from and towards the masked out tokens to be zero after the softmax, preventing them from having any influence on the resulting image representation.

---

[1] $\mathbf{s} \in [0, 1]^N$ in case of a soft selection mask.

### 3.2 Stage 1: Identifying Relevant Image Regions

To identify relevant image regions for the downstream task, we leverage the PDiscoFormer part discovery method Aniraj et al. (2024). This approach, guided solely by image-level class labels and part-shaping priors, partitions the image into $K + 1$ regions, where $K$ distinct foreground parts are identified, and the remaining region represents the background, which is discarded. The discovered parts are shared across classes. Each part is associated with a learned prototype, encouraging semantic consistency across the dataset. The prototypes are also trained to be mutually de-correlated, so that each part captures a distinct aspect of the object. To this end, we use the original PDiscoFormer default settings.

### 3.3 Stage 2: Masked-input classification

PDiscoFormer suffers from the same issues that we have identified as flaws in attention mechanisms: it uses soft attention masks that are applied to a high-level representation. To address this drawback, we propose to make the masks binary, via discretization, and to use them to explicitly define the receptive field of the second stage model, using Eq. (4).

**Discrete masks.** PDiscoFormer produces part attention maps that assign, for each image token, a weight distribution across parts, with weights summing to one. These weights are designed to approach a hard assignment via Gumbel softmax, where one part receives a weight close to one, while the others are close to zero. However, we emphasize that these maps still remain a soft distribution across parts. This may seem as a subtlety, but we argue that only a truly discrete attribution map can provide faithfulness guarantees by fully preventing information leakage. To tackle this issue, we introduce a discretization step for the obtained part maps prior to the second stage. At this point, the foreground parts are merged together to obtain a binary input mask for the second stage model. With the aim to allow gradient flow between the second and first stages, we employ the straight-through gradient trick used by Gumbel softmax Jang et al. (2017), where the hard masks are used in the forward pass and the soft ones in the backward pass.

**Input image masks.** An additional requirement in order to prevent information leakage, related to the receptive fields of modern computer vision architectures, is to adopt early masking Aniraj et al. (2023). That is, masking directly the input of the model instead of doing so at a higher-level representation. In this way, only the unmasked tokens are considered by the ViT, thus eliminating any possible information contamination from the unattended regions. To mitigate potential impacts on training dynamics from removing background tokens, our ViT architecture also incorporates register tokens, which are restricted to attend only to the foreground.

**Part dropout.** During training, we randomly drop out discovered image parts with a probability $p$. This not only helps to promote robustness to missing parts in the second stage (which will be useful for the intervention functionality discussed in Sec. 3.4), but also makes sure that all parts have the opportunities to backpropagate useful learning signals to the first stage, as the stage-2 model cannot always rely on a single informative part to perform classification.

### 3.4 Test-time Correction/Interventions

Although the stage-1 training objective encourages foreground discovery, spurious objects or correlations may still be captured due to the weakly supervised nature of the task. Unlike standard DL models, our framework is locally interpretable, meaning it faithfully reveals the image regions responsible for solving the task. This property enables targeted test-time corrections to mitigate learned spurious correlations. Here, we propose two intervention methods.

**Drop a part that captures a spurious object.** The original PDiscoFormer, due to the asymmetry in the treatment of the background part, exhibits a bias toward assigning as much as possible of the image content to the background, the unattended image regions. This implies that the discovered parts are typically the most informative for the downstream task, often corresponding to the image regions that are causally related to the classification label. However, when the number of parts $K$ is set sufficiently high, some parts may begin to focus on spurious correlations. iFAM allows the users to *select, at inference time, a subset of*

*the discovered parts to feed into the stage-2 classifier.* Since the part discovery component encourages each part to capture semantically consistent content across the dataset, *this operation can be performed globally.* This allows for the manual inspection of a few images (see Appendix D) to gain insights into what each part captures. If one of the parts is found to consistently capture an element associated with a spurious correlation, it can be excluded from the input to the second stage.

**Drop tokens assigned to a part with low confidence.** In cases where OOD objects present at inference time lead to false positive part detections, it is possible to simply remove the low confidence tokens from any given part. This can be achieved by checking whether the assigned parts are unexpectedly distant from the corresponding prototype in the feature space, based on statistics drawn from the training set Liu et al. (2020). Specifically, a distance-based threshold $\tau_k^q$ can be calibrated on the training set given a large percentile $q$, such that $q$ is the proportion of tokens assigned to part $k$ that have a distance to the corresponding part prototype smaller than $\tau_k^q$. At inference, tokens assigned to part $k$ with distance exceeding $\tau_k^q$ are reclassified as background.

Finally, since these two approaches are complementary, the first addressing part-level intervention while the second covers individual tokens from all parts, they can be adopted simultaneously.

## 4 Experimental Setup

We aim to discover task-relevant image regions using only image-level class labels, applying attention masking to restrict the predictor's receptive field and focus solely on these regions. To evaluate the effectiveness of our approach, we use datasets with known background-related biases or other spurious correlations.

### 4.1 Datasets and Evaluation Metrics

We evaluate our approach on two binary classification tasks: **MetaShift cat vs. dog** Liang et al. (2022); Wu et al. (2023) and **Waterbirds** Sagawa et al. (2020), with spurious background correlations. In MetaShift, dogs predominantly appear in outdoor settings (e.g., *bench*, *bike*) and cats in indoor environments (e.g., *sofa*, *bed*) during training, while the test set contains only indoor backgrounds (e.g., *shelf*), making dogs harder to detect. In Waterbirds, derived from CUB Wah et al. (2011), species are assigned to *waterbird* and *landbird* classes with controlled background replacement. During training, 95% of waterbirds appear on water and 95% of landbirds on land, with the hardest groups thus consisting of waterbirds on land and landbirds on water. Both datasets report **average accuracy (AA)**, which can be inflated by leveraging background correlations, and **worst group accuracy (WGA)**, which measures robustness under background shifts.

We also train on **CUB** as a 200-way classification task and evaluate on **Waterbird200** (CUB with artificial backgrounds) to assess robustness in fine-grained scenarios.

Additionally, we assess our approach on **SIIM-ACR** Zawacki et al. (2019), a chest X-ray dataset for pneumothorax (collapsed lung) detection, where positive samples are often biased by visible chest tubes Saab et al. (2022); WGA is computed on a curated subset without this artifact.

Finally, we test the scalability to larger datasets on the **ImageNet-9 (IN-9) Backgrounds Challenge** Xiao et al. (2021), which allows direct evaluation of models trained on ImageNet-1K (IN-1K) Russakovsky et al. (2015) for background robustness. We focus on three IN-9 variants: **Original** (unaltered), **Mixed-Same** (same-class backgrounds), and **Mixed-Rand** (random-class backgrounds). **BG-GAP** Xiao et al. (2021) measures the accuracy drop from Mixed-Same to Mixed-Rand.

### 4.2 Baselines

We compare our method against several approaches from the literature, including late-masking-based PDiscoFormer Aniraj et al. (2024), standard CNN/ViT models, and dedicated de-biasing methods, across MetaShift, Waterbirds, CUB–Waterbirds200, SIIM-ACR, and IN-9. For MetaShift and Waterbirds, we also evaluated early and late masking techniques based on the result of a saliency-based foreground detection method Siméoni et al. (2023). For datasets with pixel-level annotations (e.g., masks or boxes), we additionally report results from models trained with this extra supervision as **upper bounds**.

Table 1: Results on MetaShift, Waterbird, ImageNet-1K (IN-1K), and IN-9 (Original: IN-9O; Mixed-Same: MS; Mixed-Rand: MR). BG-GAP = MS−MR (lower is better). Shaded rows (performance upper bounds): [†] models trained with extra supervision; [‡] larger-capacity models. $K$: number of foreground parts. LLE: Last Layer Ensemble Li et al. (2023), SWAG Singh et al. (2022), MAE He et al. (2022), ❄: Frozen backbone, ♠ : Fine-tuned backbone, ✗ : Intervention, gt: Ground Truth Masks, f: FOUND (Saliency detection) Siméoni et al. (2023), [1]: SWAG Singh et al. (2022) pre-train + LLE Li et al. (2023), [2]: MAE He et al. (2022) pre-train + LLE Li et al. (2023)

**(a) Results on Metashift and Waterbird**

| | | MetaShift | | | Waterbird | |
| Method | K | AA | WGA | K | AA | WGA |
|---|---|---|---|---|---|---|
| Early mask[gt†] | - | - | - | 1 | 99.2 | 97.2 |
| Late mask[gt†] | - | - | - | 1 | 95.7 | 84.0 |
| ResNet50 ERM Wu et al. (2023) | - | 72.9 | 62.1 | - | 97.0 | 63.7 |
| ViT-B ERM | - | 75.8 | 62.5 | - | 95.0 | 80.7 |
| ViT-B DinoV2 ❄ | - | 83.2 | 72.6 | - | 95.9 | 88.5 |
| ViT-B DinoV2 PCA Darbinyan et al. (2023) | - | - | - | - | 97.4 | 94.0 |
| ViT-B DinoV2 ♠ | - | 84.7 | 76.8 | - | 98.6 | 95.8 |
| ResNet50 MaskTune Asgari et al. (2022) | - | - | - | - | 93.0 | 86.4 |
| ResNet50 GroupDRO Sagawa et al. (2020) | - | 73.6 | 66.0 | - | 91.8 | 90.6 |
| ResNet50 DISC Wu et al. (2023) | - | 75.5 | 73.5 | - | 93.8 | 88.7 |
| PDiscoFormer Aniraj et al. (2024) | 2 | 86.9 | 81.0 | 4 | 96.0 | 87.4 |
| PDiscoFormer Aniraj et al. (2024) | 4 | 83.2 | 75.5 | 8 | 94.2 | 84.3 |
| PDiscoFormer Aniraj et al. (2024) | 8 | 88.7 | 83.6 | 16 | 95.9 | 85.1 |
| Late mask[f] Siméoni et al. (2023) | 1 | 82.3 | 73.5 | 1 | 95.3 | 83.3 |
| Early mask[f] Siméoni et al. (2023) | 1 | 84.5 | 77.1 | 1 | 98.6 | 95.2 |
| iFAM | 1 | 88.5 | 86.9 | 1 | 98.7 | 95.8 |
| iFAM | 2 | **89.1** | 86.3 | 4 | 98.7 | 96.4 |
| iFAM | 4 | 88.7 | **88.6** | 8 | **99.0** | 97.0 |
| iFAM | 8 | 84.5 | 78.8 | 16 | 98.8 | 97.0 |
| iFAM+✗ | 8 | 84.8 | 83.0 | 16 | 98.8 | **97.4** |

**(b) Results on ImageNet-9 (IN-9) Backgrounds Challenge**

| Method | K | IN-1K | IN-9O | MS | MR | BG-GAP ↓ |
|---|---|---|---|---|---|---|
| ResNet50 ERM Wightman et al. (2021) | - | 81.2 | 96.4 | 90.0 | 84.6 | 5.4 |
| ResNet-152 ERM [‡] Wightman et al. (2021) | - | 83.5 | 97.3 | 92.1 | 87.4 | 4.7 |
| ViT-B ERM Touvron et al. (2022) | - | 83.8 | 97.9 | 92.4 | 87.9 | 4.6 |
| ViT-L ERM [‡] Touvron et al. (2022) | - | 84.8 | 98.0 | 93.0 | 89.4 | 3.6 |
| ViT-B DinoV2 Darcet et al. (2024) | - | 84.6 | 98.1 | 93.1 | 87.1 | 6.0 |
| ViT-L DinoV2 [‡] Darcet et al. (2024) | - | **86.7** | 98.3 | **95.5** | 90.2 | 5.3 |
| ResNet50 MaskTune Asgari et al. (2022) | - | - | 95.6 | 91.1 | 78.6 | 12.5 |
| ResNet50 LLE Li et al. (2023) | - | 76.3 | 95.5 | 88.3 | 83.4 | 4.9 |
| ViT-B SWAG+LLE[1]Li et al. (2023) | - | 85.2 | 98.0 | 92.4 | 87.9 | 4.5 |
| ViT-B MAE+LLE[2]Li et al. (2023) | - | 83.7 | 97.4 | 92.5 | 88.3 | 4.2 |
| ViT-L MAE+LLE [‡2] Li et al. (2023) | - | 85.8 | 97.4 | 93.5 | 89.8 | 3.6 |
| PDiscoFormer Aniraj et al. (2024) | 1 | 83.3 | **98.4** | 93.9 | 88.6 | 5.3 |
| iFAM | 1 | 84.3 | 97.5 | 93.5 | 91.1 | **2.4** |
| iFAM + ✗ | 1 | 83.1 | 97.3 | 94.0 | **91.6** | **2.4** |

## 4.3 Implementation Details

All models are implemented in PyTorch. We use ViT-B Darcet et al. (2024) with publicly available DINOv2 weights Oquab et al. (2023) for initialization in all experiments, except on SIIM-ACR, where we use RAD-DINO Pérez-García et al. (2025). Training details are provided in Appendix A.

# 5 Results and Discussion

## 5.1 Results on robustness benchmarks

The results in Tables 1, and 2 demonstrate that our two-step approach, which explicitly limits the receptive field of the predictor to the discovered foreground regions, leads to significant improvements in robustness on datasets with spurious background correlations. Qualitative results are provided in Appendix D.

**Results on MetaShift and Waterbird.** Results on MetaShift and Waterbird (Tab. 1-a) highlight the advantage of using a pretrained DINOv2 backbone, as also noted by Darbinyan et al. (2023). Notably, simply fine-tuning DINOv2 surpasses all prior OOD robustness methods, while the same ViT-B pretrained on ImageNet does not, underscoring the impact of self-supervised pretraining. Additionally, early masking consistently outperforms late masking in robust accuracy, whether using ground-truth masks or saliency-based selection Siméoni et al. (2023). Our method significantly improves upon these baselines, improving WGA from 81.0% to 88.6% on MetaShift and from 94.0% to 97.0% on Waterbird—effectively halving the error. Only early masking with ground-truth segmentation surpasses our results. However, for $K = 8$ parts in MetaShift, performance drops sharply to 78.8% (from 88.6% at $K = 4$), suggesting that a larger number of parts leads the model to capture spurious regions. We posit that such errors can be corrected via test-time interventions, which we explore in the next section.

**Results on IN-9.** Tab. 1-b presents background sensitivity using the BG-GAP metric, which quantifies the accuracy difference between the Mixed-Same and Mixed-Rand variants. Surprisingly, vision transformers (ViTs) with advanced pre-training, such as DINOv2 Oquab et al. (2023); Darcet et al. (2024), perform worse than standard CNNs and ViTs trained purely on IN-1K following modern training protocols Touvron et al. (2022); Wightman et al. (2021), suggesting that such pre-training does not inherently improve background robustness. While ResNets incorporating de-biasing methods during training Li et al. (2023); Asgari et al. (2022) show minor improvements in BG-GAP, they perform significantly worse on individual IN-9 variants, and ViTs with post-pretraining de-biasing objectives Li et al. (2023) offer only marginal gains. In contrast, our **iFAM** model achieves the lowest BG-GAP of **2.4**, outperforming its baseline (PDiscoFormer) and all other models, including larger architectures like ViT-L and ResNet152, which demonstrates the gains stem from our design rather than model capacity.

**Results on CUB and Waterbird200.** Tab. 2-a shows that fine-tuning a DINOv2 ViT-B backbone does not scale well to fine-grained tasks. The fine-tuned CUB baseline underperforms its frozen counterpart on Waterbird200, despite improving by 2% in-distribution, suggesting overfitting to background cues. All late-masking models, including PDiscoFormer, stabilize around 76% on Waterbird200, indicating that background biases persist even with an oracle late mask. Our method achieves 86.2%, closely matching early-masked models from Aniraj et al. (2023), which rely on supervised segmentation masks. Despite using only self-discovered masks, our approach is within 2.5% of their fully fine-tuned model.

**Results on SIIM-ACR.** For SIIM-ACR (Tab. 2-b), training RAD-DINO or PDiscoFormer with late masking alone results in a biased model that overly relies on spurious correlations, leading to a WG AUC close to random performance. However, our method, with $K = 8$, achieves 69.0% WG AUC after interventions (up from 65.9%), approaching the 72.0% obtained with ground-truth bounding boxes, despite not using such additional annotations.

## 5.2 Additional robustness via interventions

In this experiment, we assess the impact of our intervention strategies on robustness to spurious correlations. Due to the weakly supervised nature of part discovery, our model may (i) identify spurious parts in datasets with stronger, more object-like spurious correlations (e.g., MetaShift, SIIM-ACR) or (ii) assign out-of-distribution (OOD) objects to the foreground (e.g., models trained on CUB and evaluated on Waterbird200). To address the first issue, we perform a **leave-one-out (LOO)** evaluation at inference, measuring its effect on WGA. For OOD foreground assignments, we remove unconfident tokens and evaluate classification performance. Additionally, we analyze the complementarity of these approaches by applying token removal on top of LOO for the worst-performing $K$ variant (without any intervention), where a spurious

Table 2: Results on CUB, Waterbird200 (CUB with OOD backgrounds) and SIIM-ACR. Shaded rows (performance upper bounds): [†] models trained with extra supervision . ❄ : Frozen backbone, ♨ : Fine-tuned backbone, ✖ : Intervention, AUC: Area Under the Curve, seg : Supervised Semantic Segmentation

**(a) Results on CUB and Waterbird200**

| Method | K | CUB in-distrib. | Waterbird200 OOD |
|---|---|---|---|
| Early mask[seg] [†] Aniraj et al. (2023) ❄ | 1 | 90.1 | 86.9 |
| Early mask[seg] [†] Aniraj et al. (2023) ♨ | 1 | 91.4 | 88.8 |
| Late mask[seg] [†] Aniraj et al. (2023) ❄ | 1 | 88.6 | 76.6 |
| Late mask[seg] [†] Aniraj et al. (2023) ♨ | 1 | 90.7 | 74.8 |
| ViT-B DinoV2 ❄ | - | 89.2 | 76.6 |
| ViT-B DinoV2 ♨ | - | **91.6** | 68.4 |
| PDiscoFormer Aniraj et al. (2024) | 4 | 89.1 | 76.0 |
| PDiscoFormer Aniraj et al. (2024) | 8 | 88.8 | 76.8 |
| PDiscoFormer Aniraj et al. (2024) | 16 | 88.7 | 75.8 |
| iFAM | 1 | 89.0 | 84.2 |
| iFAM | 4 | 90.1 | 86.1 |
| iFAM | 8 | 90.4 | 86.2 |
| iFAM | 16 | 90.6 | 86.2 |
| iFAM+✖ | 16 | 90.5 | **87.3** |

**(b) Results on SIIM-ACR**

| Method | K | A. AUC | WG AUC |
|---|---|---|---|
| BBox-ERM [†] Saab et al. (2022) | - | 92.4 | 72.0 |
| Segmentation-ERM [†] Saab et al. (2022) | - | 93.3 | 82.0 |
| ResNet50 Saab et al. (2022) | - | 90.9 | 45.5 |
| ResNet50 JTT Liu et al. (2021) | - | **92.6** | 55.9 |
| ResNet50 GEORGE Sohoni et al. (2020) | - | 92.0 | 63.4 |
| ViT-B RAD-DINO ❄ | - | 90.6 | 40.6 |
| ViT-B RAD-DINO ♨ | - | **92.6** | 54.3 |
| PDiscoFormer Aniraj et al. (2024) | 8 | **92.6** | 46.7 |
| iFAM | 8 | 92.1 | 65.9 |
| iFAM+✖ | 8 | 91.1 | **69.0** |

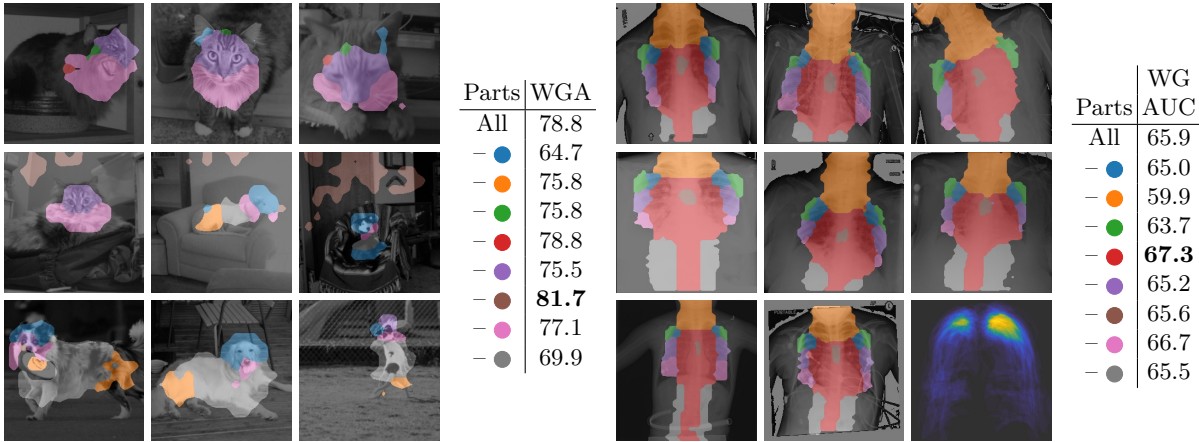

| Parts | WGA |
|---|---|
| All | 78.8 |
| − 🔵 | 64.7 |
| − 🟠 | 75.8 |
| − 🟢 | 75.8 |
| − 🔴 | 78.8 |
| − 🟣 | 75.5 |
| − 🟤 | **81.7** |
| − 🩷 | 77.1 |
| − ⚪ | 69.9 |

| Parts | WG AUC |
|---|---|
| All | 65.9 |
| − 🔵 | 65.0 |
| − 🟠 | 59.9 |
| − 🟢 | 63.7 |
| − 🔴 | **67.3** |
| − 🟣 | 65.2 |
| − 🟤 | 65.6 |
| − 🩷 | 66.7 |
| − ⚪ | 65.5 |

Figure 3: Leave-one-out (LOO) part removal intervention results on MetaShift (left) and SIIM-ACR (right) for $K = 8$. The bottom right image shows a heatmap of the average pneumothorax occurrence across the dataset.

part is likely to have been discovered, in MetaShift and SIIM-ACR. For comparison, we apply the same interventions to PDiscoFormer.

Table 3: Results of applying the token removal intervention on MetaShift, Waterbird, SIIM-ACR, and the OOD Waterbird200 dataset.

| | MetaShift (K=8) | | Waterbird (K=16) | | SIIM-ACR (K=8) | | Waterbird200 (OOD) | | |
|---|---|---|---|---|---|---|---|---|---|
| Method | AA | WGA | AA | WGA | A. AUC | WG AUC | K=4 | K=8 | K=16 |
| iFAM | 84.5 | 78.8 | **98.8** | 97.0 | 92.1 | 65.9 | 86.1 | 86.2 | 86.2 |
| ✂ $q=97\%$ | **+0.2** | +0.3 | -0.1 | -0.4 | -0.1 | +0.1 | **+0.7** | +0.5 | **+1.1** |
| ✂ $q=99\%$ | **+0.2** | **+1.3** | 0.0 | **+0.4** | **+0.1** | **+0.5** | +0.5 | **+0.7** | +0.7 |

Table 4: Results on MetaShift and SIIM-ACR using LOO and token removal, selecting the worst-performing $K$ variant without any ✂.

| | MetaShift | | SIIM-ACR | |
|---|---|---|---|---|
| Method | AA | WGA | A. AUC | WG AUC |
| PDiscoFormer Aniraj et al. (2024) | 83.2 | 75.5 | **92.6** | 48.1 |
| ✂ LOO | +2.0 | **+1.3** | 0.0 | 0.0 |
| ✂ LOO + $q=97\%$ | +2.0 | +0.3 | 0.0 | **+0.1** |
| ✂ LOO + $q=99\%$ | **+2.2** | **+1.3** | 0.0 | **+0.1** |
| iFAM | 84.5 | 78.8 | **92.1** | 65.9 |
| ✂ LOO | +0.2 | +2.9 | -1.5 | +1.4 |
| ✂ LOO + $q=97\%$ | +0.2 | +3.2 | -1.3 | +2.8 |
| ✂ LOO + $q=99\%$ | **+0.3** | **+4.2** | -1.0 | **+3.1** |

**Part-Removal Intervention on MetaShift.** Fig. 3 (left) presents part assignment maps in MetaShift, color-coded, alongside WGA results from leave-one-out (LOO) evaluation. Most parts consistently capture coherent semantics. However, the brown part is strongly biased toward indoor elements, likely due to correlations between indoor backgrounds and the *cat* class. Removing this part at inference improves WGA from 78.8% to 81.7%, whereas removing other parts either reduces performance or has no effect.

**Part-Removal Intervention on SIIM-ACR.** Fig. 3 (right) shows SIIM-ACR results, where removing the red part increases WG AUC by nearly 1.5 points. This part predominantly covers the central chest region, which has little overlap with common pneumothorax locations, as confirmed by the heatmap of average pneumothorax occurrence, but often contains spurious cues, such as drainage tubes.

**OOD Token Removal in Waterbird200.** Fig. 4 illustrates OOD token removal for $K = 8$. In CUB (second column), discovered parts align well with the bird. However, in Waterbird, background objects are often misassigned to foreground parts. Since these objects have representations farther away from part prototypes, applying a $97^{th}$ percentile threshold effectively removes them. This results in a small but consistent improvement in Waterbird200 (Tab. 3), with over a one-point gain at $K = 16$. A quantitative analysis of intervention effects on foreground and part discovery in OOD settings is provided in Appendix C.

**Combining Intervention Strategies.** Tab. 4 shows that test-time interventions provide notable gains for iFAM but only marginal improvements for PDiscoFormer. Specifically, applying both strategies improves iFAM's performance by over 4 and 3 points on MetaShift and SIIM-ACR, respectively, while PDiscoFormer sees only a 1-point and 0.1-point increase in WGA.

## 5.3 Ablation Studies

To understand the contribution of each component in our proposed method, we conduct an ablation study on the 200-way CUB/Waterbird200 benchmark and the binary MetaShift task. The results are given in Tab. 5.

**Impact of the Second Stage.** Removing the second stage of iFAM, reducing the model to PDiscoFormer, results in the steepest accuracy drop on both robustness metrics (Waterbird and MetaShift WGA). This highlights the importance of our two-stage approach in improving robustness.

**Effect of Soft Masks.** Using soft masks, where all input tokens retain some non-zero level of attention, improves in-distribution accuracy on CUB and slightly degrades performance on in-distribution MetaShift. However, it significantly reduces performance in out-of-distribution settings. This suggests that soft input

Table 5: Ablation results with $K = 4$. Rows with ** are identical.

| | CUB in-distrib. | Waterbird200 OOD | MetaShift AA | WGA |
|---|---|---|---|---|
| Full iFAM ** | 90.1 | **86.1** | **88.7** | **88.6** |
| No second stage | 89.1 | 76.0 | 83.2 | 75.5 |
| Soft masks | **90.6** | 85.7 | 88.0 | 86.3 |
| $K = 1$ w/o shaping | 90.3 | 80.2 | 85.4 | 79.1 |
| No stage-1 classif. | 88.9 | 85.0 | 86.9 | 82.3 |
| Frozen stage-2 | 89.1 | 83.7 | 85.0 | 85.0 |
| Part Dropout $= 0.5$ | 89.8 | 85.5 | 87.1 | 84.3 |
| Part Dropout $= 0.3$ ** | 90.1 | **86.1** | **88.7** | **88.6** |
| Part Dropout $= 0.1$ | 89.8 | 85.4 | 84.1 | 82.0 |
| Part Dropout $= 0.0$ | 89.9 | 85.4 | 86.5 | 86.0 |

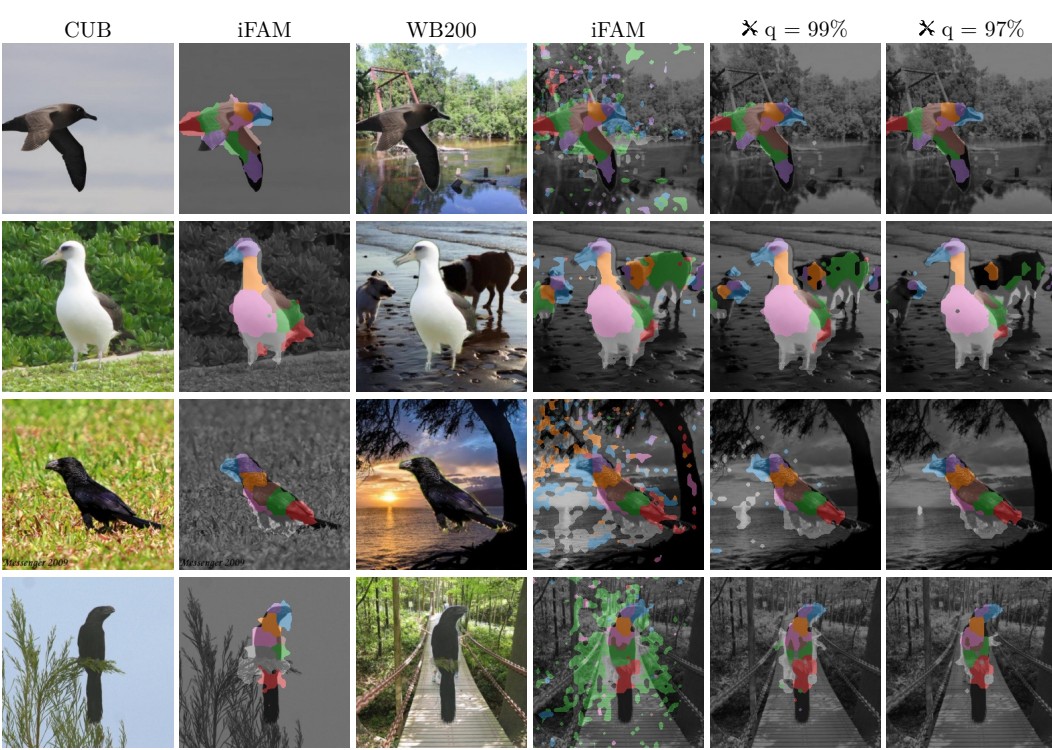

Figure 4: Qualitative results of part discovery of our model on the CUB dataset ($K = 8$), along with results on the corresponding out-of-distribution (OOD) images from the WB200 (WaterBirds200) dataset and the effect of the test-time intervention of thresholding on the OOD images.

masks allow background regions to influence stage-2 classification, leading to a weaker robustness to spurious correlations.

**Role of the first stage learning objective.** Removing only the first stage classification loss or completely removing the PDiscoFormer part discovery losses both result in notable but non-catastrophic performance drops. This suggests that, although using PDiscoFormer as stage-1 contributes to the quality of the model, the stage-2 classification is still capable to drive the foreground discovery of stage-1.

**Importance of Fine-tuning Stage-2.** Fully fine-tuning the second stage leads to consistent performance improvements, as the model cannot overfit to spurious correlations that are filtered out by stage-1.

**Part Dropout.** A sensitivity analysis on the part dropout rate in stage-2 reveals that a value of 0.3 is appropriate.

# 6 Conclusion

**Limitations.** The main limitation of our approach is the extra computational cost incurred by the use of two forward passes: one for part discovery and the second for the downstream task. While the straight-through gradient requires the entire image to be processed during training, the second pass only requires access to a subset of the image at inference, allowing optimization via patch token pruning Li et al. (2022). Additionally, our framework is currently designed for image-level tasks; adapting it for dense prediction remains an avenue for future work.

**Conclusion.** We investigated a two-step framework where stage-1 processes the full image to discover task-relevant regions, while stage-2 operates exclusively on this binary selection. By guaranteeing the receptive field of the stage-2 predictor through attention masking, we ensure that only the regions identified by stage-1 influence its representations, thereby minimizing background-related biases. Empirically, we show that this approach significantly improves robustness on benchmarks designed to test resilience against such biases. Our findings highlight the importance of inherently faithful attention mechanisms for developing robust computer vision systems.

# 7 Broader Impact Statement

Our method's architectural guarantee of faithfulness enhances transparency, which is critical in high-stakes applications. For instance, in medical imaging, it allows for auditing the model's focus to prevent reliance on spurious cues like chest tubes Saab et al. (2022). While a risk of over-filtering relevant context exists, the explicit nature of the masks makes this risk directly auditable by domain experts. This transparency also helps mitigate misuse by making it easier to detect when the model relies on inappropriate or biased features, fostering more accountable AI.

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

## A  Training Settings

We trained all models for 90 epochs using the AdamW optimizer Loshchilov & Hutter (2019). During the part discovery stage, we followed the procedure outlined in the original paper Aniraj et al. (2024). Specifically, the class token, position embedding, and register token were kept unfrozen, while the remaining ViT layers were frozen. In this stage, we trained these unfrozen tokens along with the randomly initialized layers, including the projection, modulation, and final classification layers. In the second stage, we fine-tuned all parameters of the model.

To adjust the learning rate dynamically, we employed a cosine annealing schedule Loshchilov & Hutter (2022). The initial learning rates were set as follows: $10^{-6}$ for the fine-tuned tokens of the ViT backbone in both stages and for the layers of the second-stage ViT, $10^{-3}$ for the linear projection layer forming the part prototypes, and $10^{-2}$ for the modulation and final linear layers used for classification in both stages.

We used a variable batch size, with a minimum of 16, depending on the available computational resources. To scale the learning rate appropriately, we applied the square root scaling rule Krizhevsky (2014). Regularization was performed using gradient norm clipping Pascanu et al. (2013) with a constant value of 2 and a normalized weight decay Loshchilov & Hutter (2019) set to 0.05.

The PDiscoFormer losses were configured as in the original paper Aniraj et al. (2024), with one exception for the biomedical dataset SIIM-ACR Zawacki et al. (2019). For this dataset, we disabled the background loss $\mathcal{L}_{p_0}$ by setting its weight to 0, as this loss assumes the background part is more likely to occur at the image boundaries — an assumption that does not necessarily hold for pneumothorax occurrences.

Finally, we used a constant part dropout value of 0.3 for both stages of the model in all experiments. The dropout value for the first stage aligns with that used in the original PDiscoFormer paper Aniraj et al. (2024), while the value for the second stage was ablated in Table 5 of our main paper.

**Scaling up to larger datasets.** For larger datasets such as ImageNet1K Russakovsky et al. (2015), we adopted optimizations including Automatic Mixed Precision (AMP) Micikevicius et al. (2018) and temporal averaging using Exponential Moving Average (EMA) Kingma (2015); Morales-Brotons et al. (2024) to accelerate and stabilize training. By leveraging these optimizations, we were able to double the batch size, leading to a 3.5× reduction in training time, all while maintaining performance. Additionally, we found that larger datasets benefited from longer training, prompting us to increase the total number of epochs to 120.

**Baseline Training Settings.** Wherever possible, we report results from cited papers or evaluate public weights; otherwise, we re-train baselines using the experimental setup from the original paper.

## B    Training Time and Inference Speed

We use an input image size of 518 for the CUB Wah et al. (2011), Waterbirds Sagawa et al. (2020), SIIM-ACR Zawacki et al. (2019) aligning with the default resolution of DINOV2. This higher resolution is consistent with prior works van der Klis et al. (2023); Aniraj et al. (2024); Saab et al. (2022). For the MetaShifts Liang et al. (2022) and ImageNet1K datasets, we adopt a reduced input size of 224, resulting in lower computational requirements.

**Training Time.** On a machine with 8 NVIDIA A100 GPUs, the training times are as follows: approximately 3 hours for CUB and Waterbirds, 5 hours for SIIM-ACR, 11 minutes for MetaShifts, and 34 hours for ImageNet-1K (with AMP and EMA optimizations).

**Inference Speed.** On an RTX 3090, models trained on CUB (input size: 518) run at 43 images/second, while those trained on MetaShift (input size: 224) reach 151 images/second. These results are reported without any inference-time optimizations. We believe future work can further improve speed by leveraging the sparsity of second-stage inputs.

## C    Quantitative Analysis of Token Removal

In Table 3 of our main paper, we demonstrated that the test-time intervention of OOD/Low-confidence token removal consistently improves classification accuracy for models trained on CUB when evaluated on the Out-of-Distribution dataset WaterBird200. Additionally, this technique enhances qualitative foreground object discovery, as illustrated in Figure 4 of the main paper. In this section, we provide a detailed quantitative analysis of these results, focusing on the model's part assignment consistency and foreground discovery capability under the intervention.

**Evaluation Metrics.** The CUB dataset provides ground-truth annotations for parts in the form of keypoints, which denote the centroid locations of parts within each image, as well as foreground-background masks. Since the images in the Waterbird200 dataset are identical to those in CUB, differing only in their adversarial backgrounds, the CUB annotations can also be used for Waterbird200. We evaluate foreground discovery using **mean Foreground Intersection-over-Union (Fg. mIoU)** and part assignment consistency using **Keypoint Regression (Kp)**.

Table 6: Quantitative analysis of the effect of the token removal intervention on part assignment consistency using keypoint regression (Kp) and foreground discovery (Fg. MIoU) on the OOD Waterbird200 dataset. $K$: Number of foreground parts.

| Method | K | Kp ↓ | Fg. MIoU ↑ | Top-1 Acc. ↑ |
|---|---|---|---|---|
| iFAM | | 10.3 | 63.7 | 86.1 |
| ✂ $q = 97\%$ | 4 | **8.4** | 65.2 | **86.8** |
| ✂ $q = 99\%$ | | 9.2 | **65.9** | 86.6 |
| iFAM | | 9.3 | 68.6 | 86.2 |
| ✂ $q = 97\%$ | 8 | **6.7** | 71.4 | 86.7 |
| ✂ $q = 99\%$ | | 7.3 | **72.4** | **86.9** |
| iFAM | | 8.0 | 70.2 | 86.2 |
| ✂ $q = 97\%$ | 16 | **6.2** | 72.9 | **87.3** |
| ✂ $q = 99\%$ | | 6.5 | **73.1** | 86.9 |

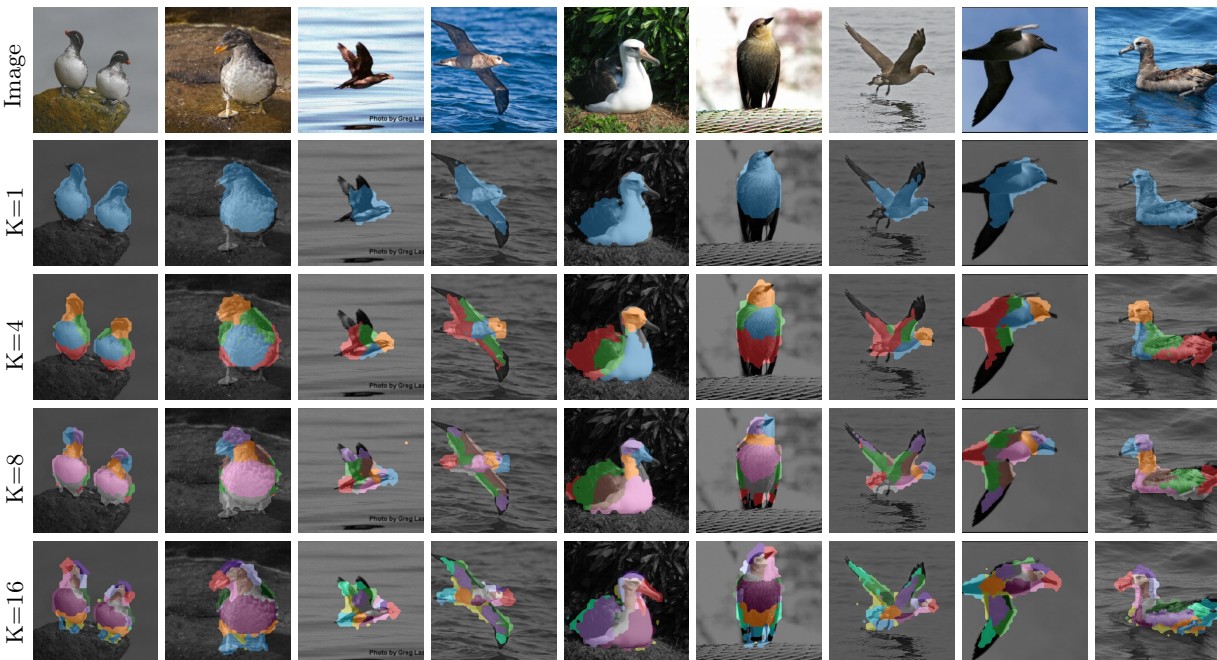

Figure 5: Qualitative results for part discovery for the iFAM model (without any ✂) trained on the CUB dataset for different values of K, the number of foreground parts.

1. **Fg mIoU.** This metric assesses the model's ability to identify the foreground region relevant for downstream classification. We merge all detected foreground parts and compute the IoU between the merged parts and the ground-truth foreground-background masks from the CUB dataset.

2. **Kp.** Following Hung et al. (2019), we measure part assignment consistency by deriving landmark locations through a trained linear regression model. This model maps the 2D geometric centers of the part assignment maps to their corresponding ground-truth part landmarks. The predicted landmarks are then compared against ground-truth annotations on the test set, with the evaluation metric being the normalized mean L2 distance.

**Results on Foreground Discovery.** The low-confidence token removal technique consistently improves Foreground MIoU across all values of $K$ on the OOD Waterbird200 dataset (see Tab. 6). However, increasing the threshold (e.g., ✂ $q{=}97\%$) leads to a slight reduction in MIoU compared to using ✂ $q{=}99\%$. For instance,

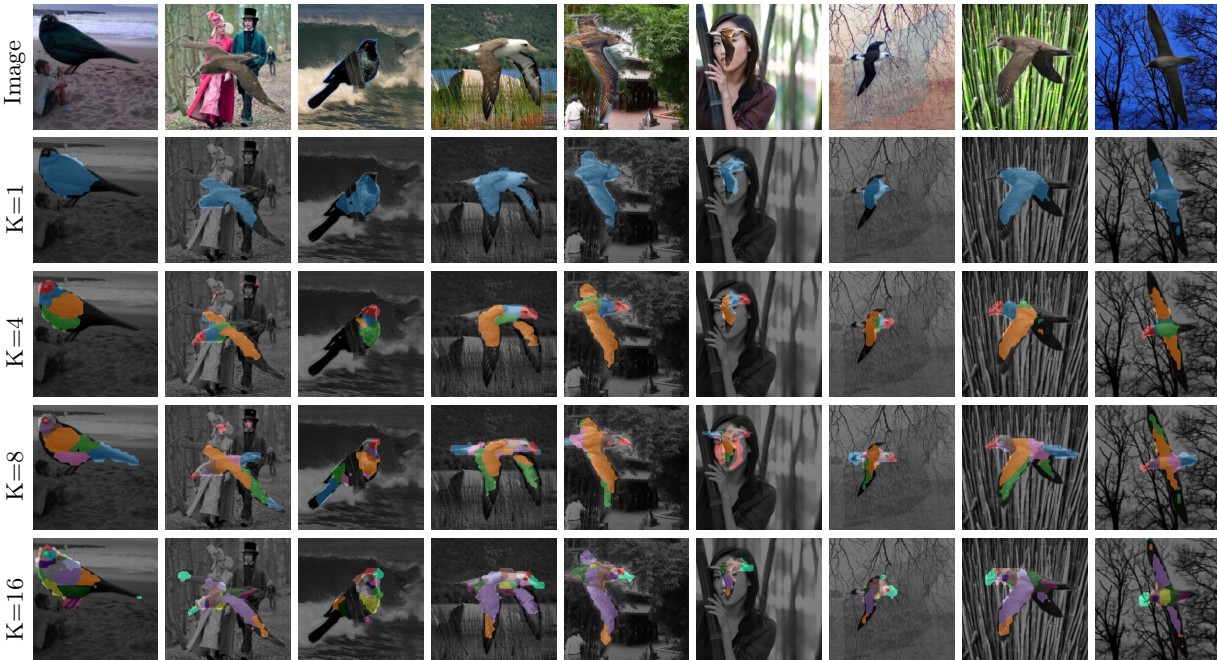

Figure 6: Qualitative results for part discovery for the iFAM model (without any ✂) trained on the Waterbirds dataset for different values of K, the number of foreground parts.

at $K = 8$ (results shown in Figure 4 of the main paper), the baseline model achieves a Foreground MIoU of 68.6%, which improves to 72.4% with ✂ $q = 99\%$, but drops to 71.4% with ✂ $q = 97\%$, suggesting that a stricter confidence threshold may inadvertently remove some foreground regions. Despite this, the drop in classification accuracy is minimal (from 86.9% to 86.7%), indicating that the model remains robust to removed foreground regions. Similar trends are observed across other values of $K$, where ✂ $q = 99\%$ generally leads to the best Foreground MIoU, while ✂ $q = 97\%$ provides slightly better classification performance.

**Results on Part Assignment Consistency.** The intervention improves keypoint regression (Kp) values across all $K$ values, indicating that the centroids of part assignment maps align more closely with ground-truth annotations. For instance, at $K = 16$, the Kp value improves from 8% (baseline) to 6.2% (✂ $q = 97\%$), likely due to the removal of low-confidence tokens near part boundaries, as shown in Fig. 4.

Overall, these results suggest that low-confidence token removal enhances both foreground discovery and part assignment consistency, with ✂ $q = 99\%$ generally yielding the best Foreground MIoU, while ✂ $q = 97\%$ slightly improves classification performance.

## D Qualitative Results for Part Discovery

To complement the quantitative evaluations in the main paper, we provide additional qualitative results in Figures 5 to 10. These results demonstrate our model's ability to discover meaningful parts and accurately identify foreground regions, which are crucial for downstream classification tasks and improving model interpretability.

**Results on CUB and WaterBird.** In datasets such as CUB and Waterbird, where all images belong to a single super-class (birds), the granularity of the discovered parts improves as $K$ increases. The identified parts generally align well with the foreground regions, as shown in Fig. 5 and Fig. 6.

**Results on MetaShifts.** For the binary classification task in MetaShifts (Cat vs. Dog), illustrated in Fig. 7, the model assigns a single part (blue) to both cats and dogs when $K = 1$. At $K = 2$, the same part (orange) is assigned to both classes, while another part (blue) is allocated to objects that frequently co-occur

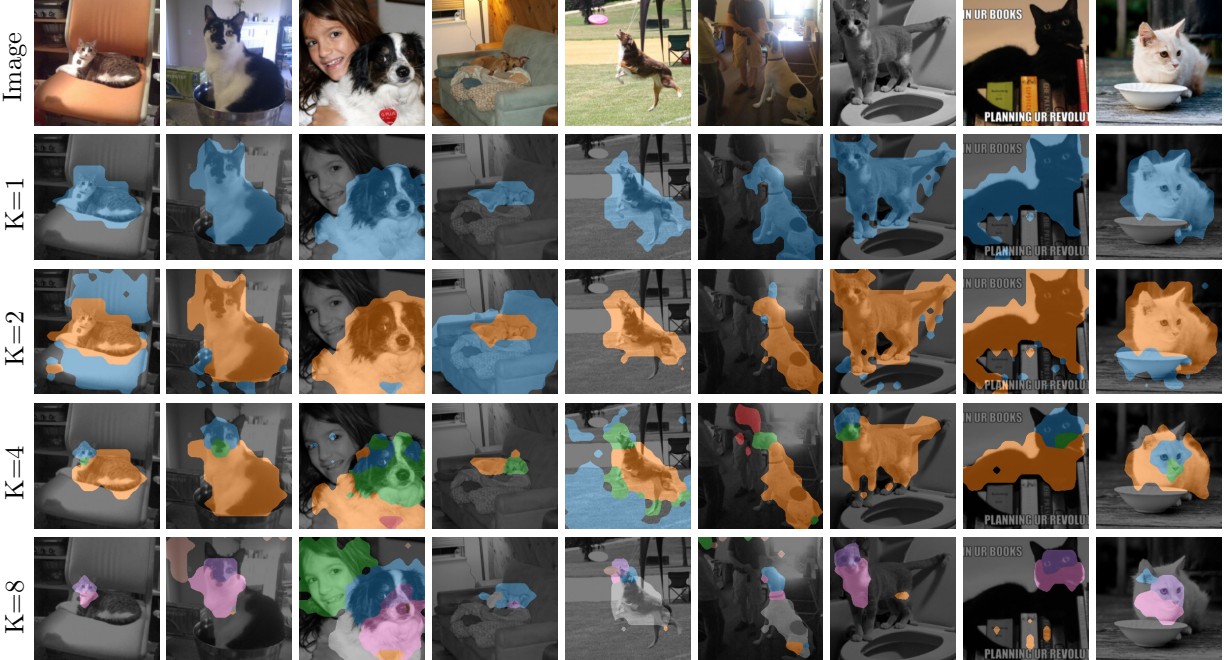

Figure 7: Qualitative results for part discovery for the iFAM model (without any ✂) trained on the MetaShifts dataset for different values of K, the number of foreground parts.

with these animals in the training set. However, at higher values of $K$, such as $K = 8$, the model begins to identify more non-causal or spurious parts, likely explaining the performance drop observed for this variant in Table 1-a of the main paper.

**Results on ImageNet-1K.** Qualitative results on ImageNet-1K for various animal classes, including birds, cats, dogs, and insects, are shown in Figures 8, 9, and 10 for $K = 1$. At this setting, the model effectively performs foreground discovery, which appears to generalize well across the 1000 classes of ImageNet. This observation aligns with our quantitative results on background robustness in Table 1-b of the main paper.

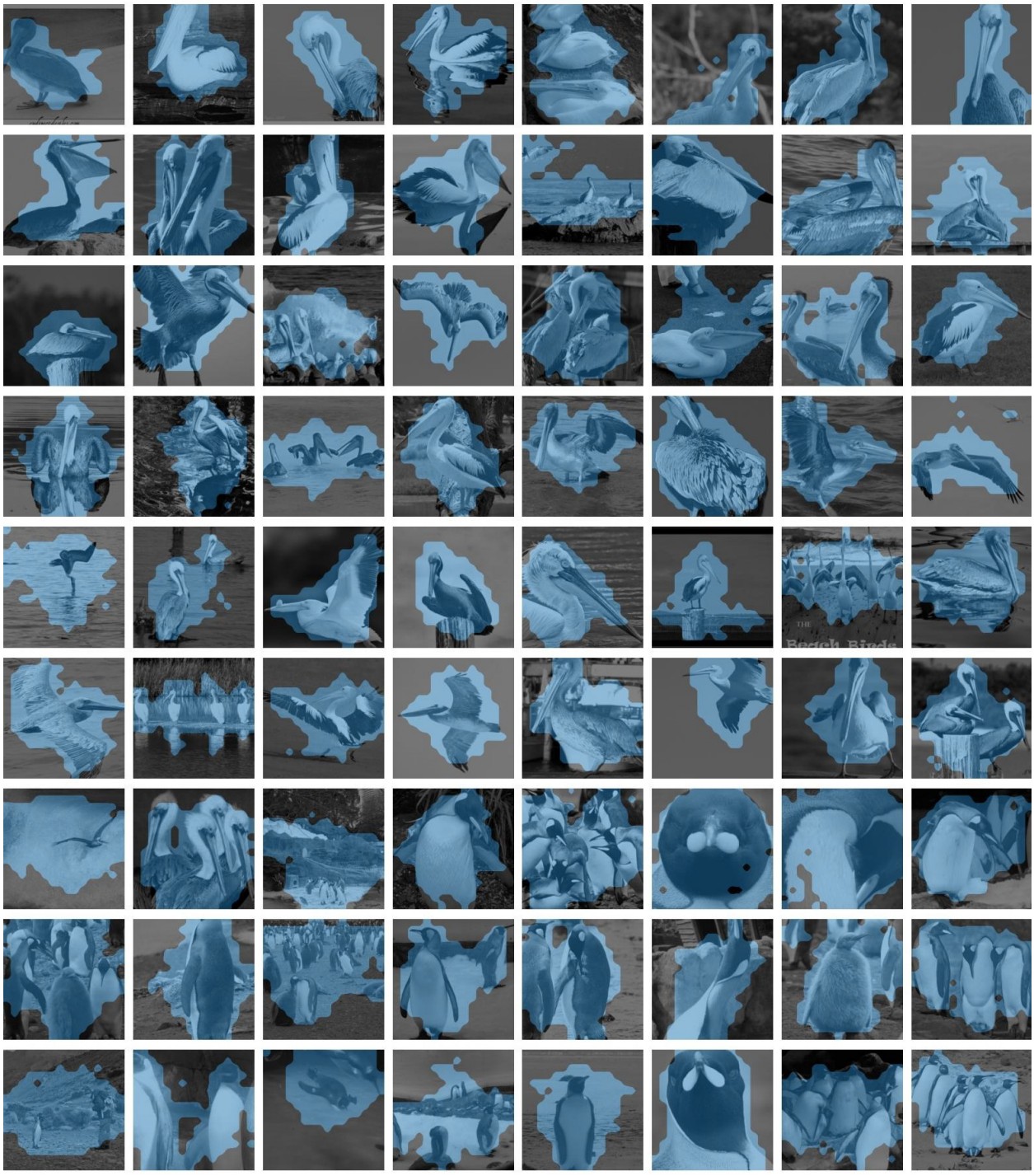

Figure 8: Qualitative Results on ImageNet-1K for Birds (without any ✗) for $K = 1$.

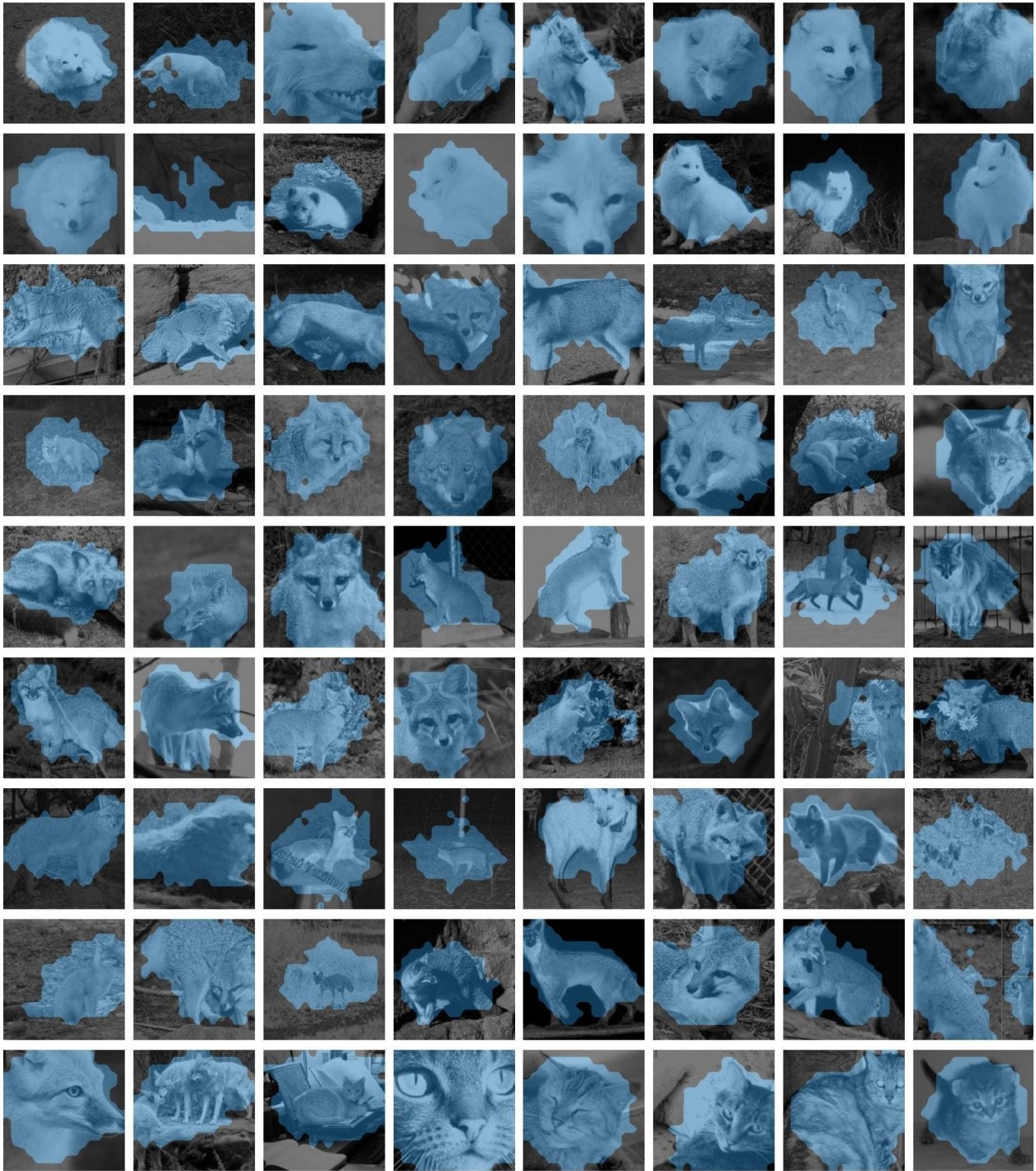

Figure 9: Qualitative Results on ImageNet-1K for Cats and Dogs (without any ✂) for $K = 1$.

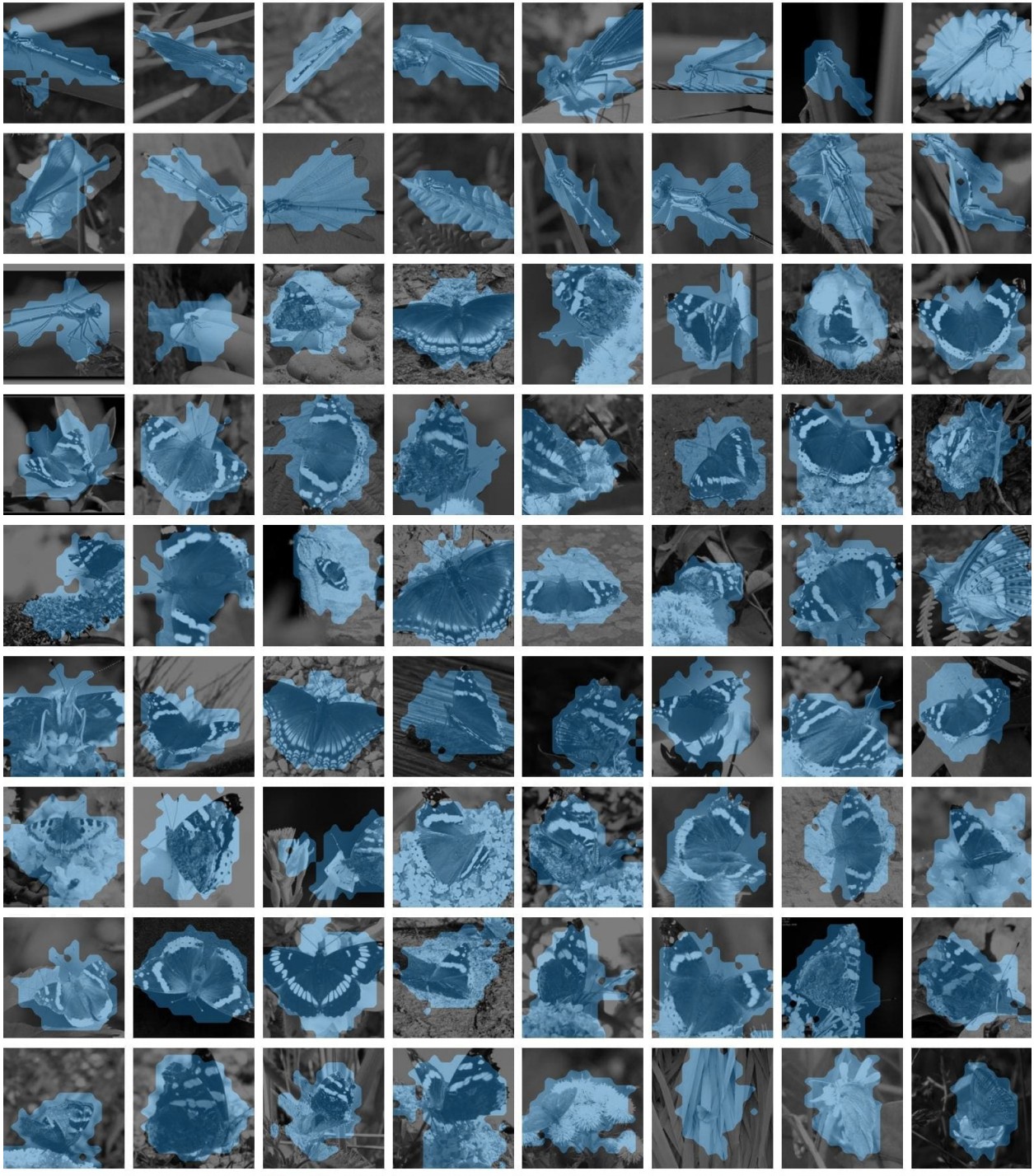

Figure 10: Qualitative Results on ImageNet-1K for Insects (without any ✂) for $K = 1$.

