# OpenReview forum: "Inherently Faithful Attention Maps for Vision Transformers"
_TMLR — Rejected by TMLR_

### Review · Reviewer_5A2K · 2025-08-03

**Summary Of Contributions:**

This paper propose to introduce learnable mask in Transformer based Neural Network classifiers. The author(s) argue that such approach could force the model focus on the critical regions and thus avoid spurious correlations.

**Audience:**

Yes

**Claims And Evidence:**

No

**Requested Changes:**

See weakness. My first recommendation to this manuscript is to resubmitted to computer vision venue.

**Strengths And Weaknesses:**

Pros:
The paper is easy to understand.

Cons:
1. This paper suits more for the computer vision rather than the machine learning venue. With the main contribution on adjust Transformer structure, without theoretical analysis or justification of the approach.
2. In the experiments, comparing only for binary classification tasks is not convincing enough, at least some more challenging settings should be tested.
3. The proposed approach looks really simple. If it is simple and effective, at least some theoretical justification should be provided or extensive experiments are required. A simple approach tested on simple tasks without theoretical analysis is not convincing to me.

---

### Review · Reviewer_x8ub · 2025-08-08

**Summary Of Contributions:**

This paper proposes a two-stage attention-based framework for Vision Transformers (ViTs) that aims to improve their robustness and interpretability. The first stage processes the full image to identify object parts and task-relevant regions. The second stage restricts its receptive field using learned binary attention masks, focusing only on the identified regions. Both stages are trained jointly. The method is evaluated across five image classification datasets, including MetaShift, Waterbirds, CUB-Waterbirds200, SIIM-ACR, and ImageNet-9, demonstrating improved robustness to spurious correlations and out-of-distribution (OOD) backgrounds.

**Audience:**

Yes

**Broader Impact Concerns:**

The use of binary attention masks could inadvertently suppress context that is semantically relevant in certain domains (e.g., medical imaging or autonomous driving). A broader impact statement should address:

* risks of over-filtering context.
* potential misuse in sensitive applications.
* transparency in decision-making for high-stakes tasks.

**Claims And Evidence:**

No

**Requested Changes:**

* Add metrics or qualitative comparisons to support claims of improved interpretability.
* Include discussion on sparse attention and token pruning methods.
* Include comparisons with baseline ViTs under the same parameters and FLOPs budget.
* Discuss potential extensions to non-classification tasks, even if not experimentally validated.

**Strengths And Weaknesses:**

## Strengths

* The proposed method is simple and intuitive.
* The paper is well-written and scoped appropriately for TMLR

## Weaknesses

* Unclear evaluation on interpretability. The paper claims improved interpretability, but lacks quantitative metrics or user studies to validate this.
* Lack of discussion on sparse attention and token pruning. This paper is closely related to sparse attention (such as [1][2]) and token pruning methods (such as [3][4]), but they are not discussed or compared.
* Lack of in-depth analysis on the source of improvement. The improved performance may come from an increased number of parameters instead of the two-stage design.
* Limited task applicability. It seems that the proposed method is only applicable to classification tasks, but inapplicable to dense prediction tasks such as object detection and semantic segmentation.

[1] Sparsifiner: Learning Sparse Instance-Dependent Attention for Efficient Vision Transformers, CVPR 2023.
[2] BiFormer: Vision Transformer with Bi-Level Routing Attention, CVPR 2023.
[3] Dynamic Token Pruning in Plain Vision Transformers for Semantic Segmentation, ICCV 2023.
[4] DynamicViT: Efficient Vision Transformers with Dynamic Token Sparsification, NeurIPS 2021.

---

> ### Author Response · Authors · 2025-08-15
> **Response to reviewer x8ub - Weaknesses**
>
> Dear reviewer x8ub,
>
> We would like to thank you for reviewing our manuscript and for providing some useful suggestions to improve it. Below we will respond to each of the comments.
>
> **Comment**: Unclear evaluation on interpretability. The paper claims improved interpretability, but lacks quantitative metrics or user studies to validate this.
>
> **Response**: We do not claim to address all aspects of interpretability, such as human alignment or subjective usefulness, which are often assessed via user studies or quantitative proxies. Our focus is on *faithfulness by design*: Stage-2 predictions are computed exclusively from tokens within the binary mask generated by Stage-1, making it impossible for information outside the mask to influence the output. We show through OOD robustness experiments (e.g., MetaShift, Waterbirds, SIIM-ACR, IN-9) that such faithful-by-design approaches effectively prevent the model from relying on background information, meaning that the masks are faithful, and we provide qualitative visualizations in Appendix D. While this property has clear implications for interpretability, a deeper exploration of these aspects lies outside the scope of this work.
>
> **Comment**: Lack of discussion on sparse attention and token pruning. This paper is closely related to sparse attention and token pruning methods, but they are not discussed or compared.
>
> **Response**: Sparse attention methods generally select tokens based solely on task discriminativeness at earlier stages of the network and sparsify the attention map (e.g., via thresholding) to reduce the quadratic complexity of attention. Similarly, token pruning methods remove tokens to accelerate computation. In contrast, iFAM leverages global image context and optimizes a joint objective combining classification losses (at both stages) with part-shaping losses that encourage spatial consistency, ensuring that spatially adjacent, semantically related tokens are activated or suppressed together. This allows iFAM to reliably distinguish background from meaningful foreground parts.
> If sparse attention or token pruning were applied in a two-stage manner without these part-shaping constraints, they would resemble our K=1 ablation (without part shaping losses) and could suffer from the same failure modes as Joint Amortized Models (see Related work: Input attention maps for interpretability). While sparse attention and token pruning primarily address efficiency, our approach targets faithfulness and robustness; the methods are orthogonal but could be combined to further accelerate the Stage-2 ViT. This is already mentioned in the **Limitations** section but we can add a further discussion in the related work.
>
> **Comment**: Lack of in-depth analysis on the source of improvement. The improved performance may come from an increased number of parameters instead of the two-stage design.
>
> **Response**: In our ImageNet-9 experiment (Table 1-b), we compare iFAM (ViT-B backbone) against several larger-capacity baselines, including ViT-L and ResNet-152 models (shaded rows). While these methods do show a higher classification accuracy on tasks with in-distribution image backgrounds such as the ImageNet (IN-1K) validation set, the original test set of IN-9 and the Mixed-Same (MS) variant of IN-9 , they are still outperformed by iFAM on the Mixed-Rand (MR) dataset wherein foreground-background correlations are mitigated by mixing random backgrounds with foregrounds. This robustness advantage is captured by the BG-GAP metric, where iFAM achieves the smallest drop in accuracy from MS to MR, outperforming all baselines despite using fewer parameters.
> We also include a detailed ablation study in Table 5, showing how each design choice (e.g., part shaping, two-stage masking) contributes to this robustness, independent of model capacity.
>
> **Comment**: Limited task applicability. It seems that the proposed method is only applicable to classification tasks, but inapplicable to dense prediction tasks such as object detection and semantic segmentation.
>
> **Response**: Our approach is primarily designed for image-level tasks, such as classification, regression, and retrieval. Pixel-level tasks like semantic segmentation fall outside the current scope. However, the core principle—restricting the receptive field via a learned binary mask—could, in principle, be adapted to generate localized representations for multiple detected regions in Stage-2. We believe exploring such adaptations for dense prediction tasks is an interesting direction for future research. We will update the manuscript to incorporate this information.

---

> ### Author Response · Authors · 2025-08-15
> **Response to reviewer x8ub - Broader Impact Concerns**
>
> **Comment**: The use of binary attention masks could inadvertently suppress context that is semantically relevant in certain domains (e.g., medical imaging or autonomous driving). A broader impact statement should address:
> - risks of over-filtering context.
> - potential misuse in sensitive applications.
> - transparency in decision-making for high-stakes tasks.
>
> **Response**:
> As demonstrated in our experiments on the **SIIM-ACR** dataset for pneumothorax detection, our approach can effectively remove spurious context—such as the presence of chest tubes, which frequently co-occur with positive cases and can mislead standard CNNs or ViTs. In this scenario, suppressing such misleading background information improved robustness without discarding semantically relevant cues. We believe this property can also be beneficial in domains like **autonomous driving**, where certain background patterns (e.g., weather artifacts) could otherwise bias predictions.
> - **Transparency for high-stakes tasks** – Our method provides an *architectural guarantee* of faithfulness-by-design: Stage-2’s predictions are computed solely from the foreground regions identified by Stage-1, and these regions are explicitly available for inspection, ensuring clarity on what information influenced each decision.
> - **Risks of over-filtering context and potential misuse in sensitive applications** – Due to the inherent faithfulness of the attention maps generated by our approach, it becomes possible to verify, potentially in consultation with domain experts, whether useful context is being filtered out. This can also potentially aid with preventing misuse. We will add a broader impact statement addressing these points.
>
> Thanks for the suggestions.

---

### Review · Reviewer_xsMF · 2025-08-11

**Summary Of Contributions:**

The authors propose a masking method for attention maps in order to make downstream tasks only depend on relevant parts of the input image and prevent spurious correlations. This is achieved with a discrete masking mechanism which divides the image into mutually exclusive regions, which determine how the attention map is used for downstream tasks.

**Audience:**

Yes

**Claims And Evidence:**

No

**Requested Changes:**

The authors show experimentally that the proposed method outperforms other baselines. In addition, i would request the authors to further elaborate on their motivation about using attention masks with respect to the points above. Mainly with regards to how interpretability affects trainability and why an additional mask is the best option to do so, could there be any other regularization based methods for example which can achieve similar results.

**Strengths And Weaknesses:**

The proposed method aims to identify the relevant regions of an attention map to make better decisions in the downstream task, however, i have several questions about this problem.

1. The authors mention the attention maps must be faithful to prevent spurious correlations, but this is not clearly defined. There could be a possibility where certain regions which are not semantically relevant would still influence downstream performance.
2. While the goal is to make attention faithful, the authors do not describe how this affects the learning dynamics, generalization ability of the method. For example, attention maps are known to have sink tokens which aid training, how does masking affect it.
3. It is unclear whether the goal of the proposed method is to enhance interpretability via 'faithful' attention. If I understand correctly, the goal of an attention map is to identify the relevant areas in an image per token. Instead the proposed method finds a discrete attention mask before computing the soft attention? How is this proposed method more interpretable?
4. As the authors point out, identifying these discrete regions introduces additional compute overhead.

---

> ### Author Response · Authors · 2025-08-15
> **Response to reviewer xsMF - Strengths and Weaknesses**
>
> Dear reviewer xsMF,
>
> We would like to thank you for reviewing our manuscript and for providing some insightful suggestions to improve it. Below we will respond to each of the comments.
>
> **Comment:** The authors mention the attention maps must be faithful to prevent spurious correlations, but this is not clearly defined. There could be a possibility where certain regions which are not semantically relevant would still influence downstream performance.
>
> **Response:** We consider an attention map to be faithful if the image regions highlighted by the attention map are actually important for the task. This is typically measured via pixel removal [A]. Soft attention applied on the last layer representation of the model, as commonly used in the literature, can be unfaithful due to two distinct issues:
> Due to being soft, all image regions tend to receive a non-zero weight. This means that, in principle, even low attention regions may influence the output in a non-negligible way.
> In addition, the large receptive fields in later layers of CNNs or ViTs mean that a local representation of an image may contain information from any other region, leading to another potential cause of unfaithfulness when masking deep image features. To address this, we apply an early binary mask strategy at the input level of the second stage, which strictly limits the receptive field to the selected foreground regions. This guarantees that no information outside the mask can influence the prediction and is equivalent to the pixel removal strategy used to measure faithfulness, leading to perfect faithfulness by design.
> It is true that the weakly supervised segmentation used in Stage-1 may sometimes include regions that are not semantically relevant. However, such regions are explicitly visible in the derived foreground mask, making it transparent which parts of the image can be used for the decision. This visibility is a key part of our definition of faithfulness, and we propose intervention strategies to remove or adjust such regions when necessary.
>
> We will better clarify this in the manuscript, and would be happy to further suggestions from the reviewer to ensure that the message is clear.
>
> [A] Wu, Junyi, et al. "On the faithfulness of vision transformer explanations." Proceedings of the IEEE/CVF Conference on Computer Vision and Pattern Recognition. 2024.
>
> **Comment:** While the goal is to make attention faithful, the authors do not describe how this affects the learning dynamics, generalization ability of the method. For example, attention maps are known to have sink tokens which aid training, how does masking affect it.
>
> **Response:** It is true that removing background tokens may have an effect on training dynamics, since they become unavailable for the model to be used for auxiliary computations, akin to sink tokens in transformer-based models. To mitigate this, we employ register tokens, which have been shown to reduce the emergence of sink tokens in Vision Transformers. In iFAM, these register tokens are restricted to attend only to the detected foreground regions. In the paper, we focus on measuring generalization (both in-distribution and out-of-distribution) after convergence, rather than on exploring the effect of masking on training dynamics.
>
> **Comment:** It is unclear whether the goal of the proposed method is to enhance interpretability via 'faithful' attention. If I understand correctly, the goal of an attention map is to identify the relevant areas in an image per token. Instead the proposed method finds a discrete attention mask before computing the soft attention? How is this proposed method more interpretable?
>
> **Response:** Our goal is to demonstrate the need for a two-stage approach, combined with the use of discrete masks on the input image before the second stage, in order to preserve the faithfulness of the mask. We believe this carries an important message for the use of attention maps for interpretability, but interpretability per se is not the focus of this paper. Instead, we evaluate faithfulness via the impact it has on OOD background generalization.
>
> We would like to clarify that, after the discrete mask is applied to the input image before the second stage, the masked out image regions become unavailable to the model. This means that, no matter what the soft attention in the second stage ViT does, we know with certainty that these tokens won't be used, which is where the inherent faithfulness comes from. Within the available foreground tokens we cannot know for sure, as we then deal again with the issues stemming from soft attention, but we keep the guarantee that the regions masked out by the discrete mask are not at all used in the computations of the second stage. This inherent faithfulness leads to improved OOD performance in the presence of spurious correlations. Being inherently faithful would have advantages for interpretability, which we plan to explore in future work.

---

> ### Author Response · Authors · 2025-08-15
> **Response to reviewer xsMF - Requested Changes**
>
> **Comment**: In addition, i would request the authors to further elaborate on their motivation about using attention masks with respect to the points above. Mainly with regards to how interpretability affects trainability and why an additional mask is the best option to do so, could there be any other regularization based methods for example which can achieve similar results.
>
> **Response**:  We believe that a two-stage approach with a discretized spatial mask fed to the second stage is a conceptually effective way to guarantee control over the information used for the downstream task.
> Regularization-based methods, such as those involving intermediate tasks like concept discovery or part-prototype learning, can indeed encourage models to focus on semantically relevant regions. However, to the best of our knowledge, these approaches still rely on late masking, where irrelevant regions are suppressed only at deeper feature levels. As evaluated and discussed in our paper, late masking does not fully prevent information leakage from background regions due to large receptive fields. This can be quantitatively measured via OOD generalization in the presence of spurious correlations. By contrast, our early, binary masking at the input of the second stage ensures that no information from outside the selected regions can influence the prediction.
>
> We will make updates to the manuscript to make the motivation more clear.

---

### Author Response · Authors · 2025-08-07
**Response to Reviewer 5A2K**

Dear reviewer 5A2K,

We would like to thank you for reviewing our manuscript. We would like to clarify, and ask for some additional clarity, on the remarks:


1. Our work indeed pertains to machine learning models for computer vision, making it fall within the scope of TMLR. We appreciate the reviewer's suggestion, but we have weighted the different option and decided that TMLR was the right choice. With respect to the justification, we clearly motivate both the importance of the problem and the approach.
2. We provide a very extensive experimental setup, including a variety of tasks. We focus on classification in part due to the benchmarks available, and also to bound the scope of the paper.  As described in Section 4.1, we evaluate our approach on five datasets: MetaShift Cat vs. Dog, Waterbirds, CUB (IID) evaluated on Waterbirds200 (OOD), SIIM-ACR (medical imaging), and the ImageNet-9 (IN-9) backgrounds challenge. Among these, MetaShift, Waterbirds, and SIIM-ACR are binary classification tasks. The CUB-Waterbirds200 setup evaluates robustness in a fine-grained 200-class setting, while IN-9 provides a large-scale benchmark for background robustness using a coarse-grained 9-class subset of ImageNet. Additionally, Section 5.3 includes a detailed ablation study supporting our design choices.
3. We do strive both for simplicity and effectiveness. We do provide very extensive experimental evidence about the latter.

We would be happy to further discuss any issues related to the correctness of the method and interest to the audience, and would also be happy to update the paper with any specific suggestions.

---

### Author Response · Authors · 2025-08-15
**General Response**

We would like to thank all reviewers for their effort and time spent assessing our manuscript. We have tried to respond to the raised questions in a response to each reviewer individually.

There is one issue raised by most reviewers that we feel is particularly important to clarify: in this work, we evaluate faithfulness via OOD generalization in datasets with spurious correlations that are linked to an image region (most often the background), and show that a two-stage approach with discrete masks is advantageous in this setting due to the guarantees it provides with respect to faithfulness (see response to reviewer xsMF for more details on this). We strongly believe that our results on faithfulness do have repercussions for attention-based interpretability, but we do not focus on these and leave them for future work.  We will revise the manuscript to better clarify the aim and scope.

We would be happy to hear from the reviewers if we have satisfactorily interpreted and responded to their concerns.

---

### Author Response · Authors · 2025-09-02
**Revised Manuscript**

Dear Reviewers,

Following your insightful feedback, we have now revised the submitted manuscript. Please note, all revisions in the new version of the paper are marked in **red** for your convenience. Upon acceptance, we will revert the text color to black.

We would like to thank you for your detailed feedback.

The following is a brief overview of the main changes in the revised manuscript:

- **Sharpened Framing and Motivation:**
  We have revised the Abstract and Introduction to more clearly define *inherent faithfulness* and to emphasize the paper's primary focus on improving robustness, as suggested. (Reviewer **xsMF**)

- **Expanded Related Work:**
  We have added a new paragraph to the Related Works section to discuss and distinguish our method from efficiency-focused approaches like sparse attention and token pruning. (Reviewer **x8ub**)

- **Strengthened Results Discussion:**
  We have explicitly highlighted in the Results section that our performance gains are not due to increased model capacity, making this point more prominent. (Reviewer **x8ub**)

- **Added Broader Impact Statement:**
  A new section has been added after the conclusion to discuss the potential risks, benefits, and applications of our work in high-stakes domains. (Reviewer **x8ub**)

- **Clarified Methodology:**
  We have added a brief clarification regarding the potential impact of token masking on training dynamics and our use of register tokens. (Reviewer **xsMF**)

- **Acknowledged Task Applicability:**
  We have updated the Limitations section to note that the current framework is designed for image-level tasks. (Reviewer **x8ub**)

We have aimed to address all of your concerns and have incorporated the requested changes as marked in the manuscript. We believe the paper is now improved and clearer, and we look forward to any further discussion or feedback.

---

### Decision · Action_Editor_JJSj · 2025-09-17

**Recommendation:** Reject

**Additional Comments:**

This paper presents a two-stage attention-based framework for vision transformers, where the first stage identifies task-relevant regions and the second stage performs the downstream task using only these regions. The two stages are trained jointly. By allowing only task-relevant regions to influence prediction, the method is designed to be more robust to spurious correlations and out-of-distribution backgrounds. This approach is demonstrated on image classification and utilizes a part discovery method (Aniraj et al. 2024) to identify relevant image regions.

All three reviewers lean to reject the paper. As mentioned above, it is felt that the two-stage approach to attention was not adequately justified. A decision was thus made not to accept the paper, but we hope that the review comments will be helpful in improving this work.

**Audience:**

Yes

**Audience Explanation:**

The findings of this paper may be of interest to some individuals in the TMLR audience.

**Claims And Evidence:**

No

**Claims Explanation:**

This work advocates for attention maps to be inherently faithful, such that only image regions semantically relevant to the downstream task are able to influence the output. The authors argue that other approaches for focusing attention on relevant regions may lack faithfulness due to large receptive fields. However, it is felt that the motivation for the proposed two-stage approach to attention remains unconvincing despite efforts to address this during the discussion period. Though its discrete masking may possibly have implications for interpretability, this aspect was not investigated in the paper. All three reviewers indicated that the claims were not sufficiently supported, and they all favor rejection.

---

> ### Author Response · Authors · 2025-09-22
>
> Dear action editor,
> Thanks a lot for handling our manuscript. Since we are not able to find any reaction of the reviewers to our response, is it possible that their response or the discussion between reviewers has remained accidentally private?
> It is not clear for us which elements of our claims are not supported by our experiments, and thus have no pointers towards improving the manuscript, since we are not able to find the consolidated reviewer assessment.

---

> > ### Comment · Action_Editor_JJSj · 2025-09-23
> >
> > Dear authors,
> >
> > The reviewers unfortunately did not participate in discussion. Reviewer xsMF believes the motivation for the proposed two-stage approach to attention was not convincingly justified in the paper or author response. This was expressed in a private section of their final recommendation but should have been said in the public comment instead. Reviewer 5A2K still feels after the author response that further theoretical justification and more extensive experimentation is needed. Reviewer x8ub did not elaborate in their final recommendation. Hopefully these comments will be of some use to you.

---

> > > ### Author Response · Authors · 2025-09-23
> > >
> > > Thanks for your very prompt reply and for your efforts. I hope you understand that we are a bit disappointed with this review process, since it remains unclear to us in what sense our motivation is "not justified", what kind of theoretical justification could be useful in this work nor what additional experiments would be needed. We would have liked to, at least, see the position of the reviewers with respect to our response. All of this means that we can barely use the provided feedback to improve our paper. We would like, nonetheless, to thank you and the reviewers again for handling and reviewing our manuscript.